# OtoWorld: Towards Learning to Separate by Learning to Move

**Omkar Ranadive** [* 1]    **Grant Gasser** [* 1]    **David Terpay** [* 1]    **Prem Seetharaman** [1]

## Abstract

We present OtoWorld[1], an interactive environment in which agents must learn to listen in order to solve navigational tasks. The purpose of OtoWorld is to facilitate reinforcement learning research in computer audition, where agents must learn to listen to the world around them to navigate. OtoWorld is built on three open source libraries: OpenAI Gym for environment and agent interaction, PyRoomAcoustics for ray-tracing and acoustics simulation, and *nussl* for training deep computer audition models. OtoWorld is the audio analogue of GridWorld, a simple navigation game. OtoWorld can be easily extended to more complex environments and games. To solve one episode of OtoWorld, an agent must move towards each sounding source in the auditory scene and "turn it off". The agent receives no other input than the current sound of the room. The sources are placed randomly within the room and can vary in number. The agent receives a reward for turning off a source. We present preliminary results on the ability of agents to win at OtoWorld. OtoWorld is open-source and available.[23]

## 1. Introduction

Computer audition is the study of how computers can organize and parse complex auditory scenes. A core problem in computer audition is source separation, the act of isolating audio signals produced by each source when given a mixture of audio signals. Examples include isolating a single speaker in a crowd, or singing vocals from musical accompaniment in a song. Source separation can be an integral part of solving several computer audition tasks, such

---
[*]Equal contribution [1]Northwestern University. Correspondence to: Omkar Ranadive <omkar.ranadive@u.northwestern.edu>.

*Published at the workshop on Self-supervision in Audio and Speech at the $37^{th}$ International Conference on Machine Learning*, Vienna, Austria. Copyright 2020 by the author(s).

[1]"oto": a prefix meaning "ear", pronounced *'oh-doh'*.

[2]https://github.com/pseeth/otoworld

[3]This work has made use of the Mystic NSF-funded infrastructure at Illinois Institute of Technology, NSF award CRI-1730689.

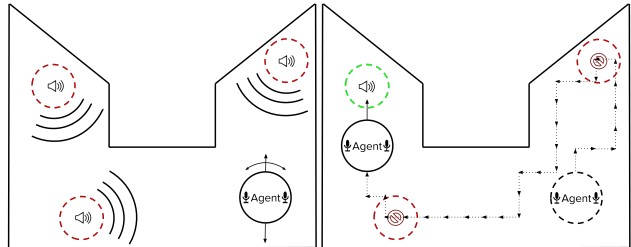

Figure 1. A visualization of the OtoWorld game. Sources are randomly placed in a room. The agent is equipped with two microphones spaced 20 cm apart to mimic human ears. The agent can navigate the environment by rotating and moving forward or backward. The left image could represent the beginning of an episode. In the right image, the agent can be seen achieving its goal of turning off sources.

as multi-speaker speech recognition (Seki et al., 2018), music transcription (Manilow et al., 2020) and sound event detection in complex environments (Wisdom et al., 2020).

The current state-of-the-art for source separation is to train deep neural networks, which learn via thousands of synthetic mixtures of isolated recordings of individual sounds. As the ground truth isolated sources that go into each mixture are known, the deep network can take as input a representation of the mixture and produce estimates of the constituent sources which can be compared to the ground truth sources via a loss function. The model can then be used to separate sources in new environments.

The way that deep models learn is very different from how humans learn to parse the auditory scene. Humans rarely have sounds presented in isolation and paired with the associated mixture. Instead, we learn directly from the complex mixtures that surround us every day (Bregman, 1994; McDermott et al., 2011a;b). Part of our ability to do this stems from our ability to interact with our environment. We can move around the world, investigate sounds, and effectively change the way we hear the world by re-positioning and orienting ourselves within it. If an unfamiliar sounds occurs in another room, we tend to navigate to the source of the unfamiliar sound and investigate what it is. By doing this, we can slowly learn about different types of sounds in our environment, and remember what they are later on.

In this work, we present a simple game environment in which researchers can explore how to create machines that can mimic this remarkable human behavior. In OtoWorld, an agent is tasked with moving around an auditory environment with only sound to guide it. The agent is equipped with two "ears", spaced 20 cm apart, which are used to listen to the environment. Multiple sounding sources are placed around the agent, which can be arbitrary in type. OtoWorld is designed to support moving or stationary sources. Several games can be designed within the OtoWorld framework. The game we design is one where the agent is tasked with discovering each source by navigating towards it. When the agent reaches a source, the source is turned off, simplifying the auditory scene and rewarding the agent. Once every source is turned off in the auditory scene, the agent has won. This game is very simple, but requires the agent to understand its auditory scene in order to win. As such, OtoWorld is a very difficult game for reinforcement learning models, as multiple tasks need to be solved - audio source classification, localization, and separation. We believe OtoWorld could be an important game to solve for training autonomous computer audition agents that can learn to separate by learning to move. An overview of OtoWorld can be seen in Figure 1.

## 2. Related Work

OtoWorld is at the cross-section of self-supervised computer audition and reinforcement learning. In the former, the goal is to learn salient audio representations that are useful for audio tasks like event detection and source separation without access to labelled training data of any kind. Several techniques for self-supervised learning exist, including bootstrapping from noisy data (Seetharaman et al., 2019; Tzinis et al., 2019; Drude et al., 2019), using supervision from a pre-trained visual network (Aytar et al., 2016), using audio-visual correspondence (Cramer et al., 2019; Zhao et al., 2018; Gao et al., 2018; Owens & Efros, 2018; Arandjelović & Zisserman, 2018), and using multi-task learning (Ravanelli et al., 2020).

In the latter, the goal is to learn agents that can move around effectively in environments by observing the current state and taking actions according to a policy. Reinforcement learning is a well-studied field. Recent advances are primarily due to the use of deep neural networks as a way to model state and produce actions (Mnih et al., 2015; Foerster et al., 2017). Audio in reinforcement learning is not as well-studied as most software that exists for research in reinforcement learning has a strong focus on visual stimuli (Ramakrishnan et al., 2020; Xia et al., 2018). Existing work incorporates both audio and visual stimuli to perform navigation tasks (Gan et al., 2019; Gao et al., 2018; Chen et al., 2019). OtoWorld is unique in that it uses *only* audio to represent the world, forcing the agent to rely strongly

on computer audition to solve tasks. Gao et al. (2020) propose VisualEchoes, the most closely related work to our own. In their work, the model must learn spatial representations by interacting with its environment with sound, like a bat would. In our work, the agent makes no sound but must navigate to sounding sources. The two approaches could be complementary to one another. Further, our goal in OtoWorld is to provide software in which researchers can easily try tasks like echolocation, source localization, and audio-based navigation.

## 3. OtoWorld

The environment is built on top of OpenAI Gym (Brockman et al., 2016), PyRoomAcoustics (Scheibler et al., 2018) and the *nussl* source separation library (Manilow et al., 2018). It consists of a room that contains an agent and unique audio sources. The agent's goal is to learn to move towards each source and "turn it off." An episode of the game is won when the agent has turned off all of the sources. Our hypothesis is that if an agent can successfully learn to locate and turn off sources, it has implicitly learned to separate sources.

### 3.1. Room Creation

The rooms are created using the PyRoomAcoustics library. Users can create a simple Shoebox (rectangular) room or more complicated n-shaped polygon rooms by specifying the location of the corners of the room. Through PyRoomAcoustics, the environment supports the creation of rooms with different wall materials and energy absorption rates, temperatures, and humidity. For more details and control, we refer the reader to (Scheibler et al., 2018) and the associated documentation.

The user provides audio files to the environment to be used as sources. Additionally, users can specify the number of sources which are then spawned at random locations in the room. Every audio source in the room has the same configurable threshold radius which is used to inform the agent that it is close enough to the audio source for it to be turned off. When initially placed, the sources are spaced such that no there is overlap of the threshold radii. By default, the sources remain in place until found by the agent.

Next, the agent is randomly placed in the room outside of the threshold radius for each audio source. The user has the option to keep the agent and source locations fixed across episodes, meaning that the initial locations of the agent and source will be the same at the beginning of each episode. Seeding the environment keeps these initializations the same for different experiment runs.

## 3.2. Action Space

The agent is allowed to take one of the four discrete actions, $\mathcal{A} = \{0, 1, 2, 3\}$, which are translated as follows:

- 0: Move forward
- 1: Move backward
- 2: Rotate right x degrees (default x = 30)
- 3: Rotate left x degrees (default x = 30)

Actions 2 and 3 are cumulative in nature, i.e. the values of degrees stack up. So, if x = 30 degrees and the agent performs action number 3 twice, then the agent rotates a total of 60 degrees to the left. If the agent chooses action 0 or 1, the agent is moved forward or backward to a new location according to the step size. Remember that the agent has two microphones (representing two human ears) and thus hears sound from left mic and right mic, receiving a stereo mix of the sources as the state. Rotation is included in the action space of the agent because the orientation of the microphones can help localize sounds more accurately.

We use the $\epsilon$-greedy approach to action selection where $\epsilon$ is the probability of choosing a *random* action from $\mathcal{A}$. This is a technique used for balancing exploration and exploitation, a commonly known trade-off in reinforcement learning. A higher epsilon causes the agent to take more random actions, which means more exploration. We balance this by initializing epsilon to a fairly high number and exponentially decaying the probability of taking random actions as the agent learns during training.

## 3.3. State Representation

At each step in an episode, the environment calculates a new room impulse response (RIR) and then provides the mixture of sources to the agent as a representation of the state of the environment. The RIR is computed using the dimensions of the room, the locations of the sources in the room, the current location of the agent, and other features of the room such as absorption. The 2-channel time-domain audio data is then returned to the agent from the step function along with the appropriate reward and the "won" flag.

## 3.4. Reward Structure

The agent is given a large positive reward (+100) for turning off a source in addition to a small step penalty (-0.5) for every action taken. There is an option to introduce dense rewards where the agent is rewarded based on its euclidean distance to the nearest source at each step. We consider the agent to have found the source once the agent is within the specified threshold radius. The source is then removed from the room environment, indicating the agent has found and "turned off" the source.

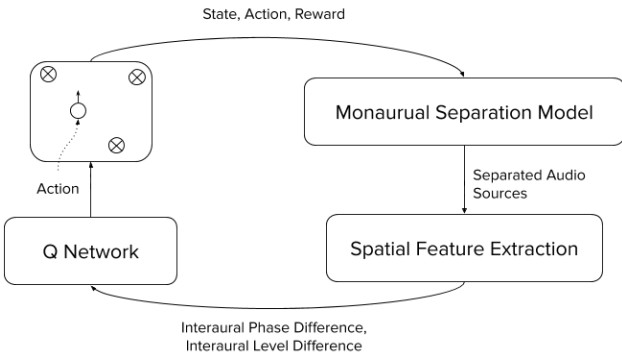

*Figure 2.* Our approach has three components - a monaural separation model for separating audio sources, a spatial feature extractor to localize the sources, and a Q-network for choosing actions.

## 3.5. Experience Replay Buffer

We employ the *experience replay* (Foerster et al., 2017) technique where the experiences of the agent are stored at each time step, $e_t = (s_t, a_t, r_t, s_{t+1})$ where $s_t$ is the current state, $a_t$ is the action, $r_t$ is the reward and $s_{t+1}$ is the next state experienced after taking action $a_t$. These observations are stored in a replay buffer data set and are randomly sampled when fed to the model.

## 4. Approach

Our approach consists of three components, as shown in Figure 2. The first component is a monaural separation network whose goal is to take the 2-channel audio representation of the state and process each channel independently to produce separated sources. The second component uses the separated sources produced by the monaural separation network to extract spatial features from the state. Finally, the last component uses the extracted spatial features along with an optional additional state such as the location of the agent and its current orientation to produce an action. The agent takes the action, and the process repeats.

## 4.1. Monaural Separation Model

The monaural separation model is a standard deep separation network based on recurrent layers (Wang et al., 2018; Luo et al., 2017; Hershey et al., 2016). The input to the network is time-domain single-channel audio. The network then takes the STFT of the audio, with a filter length of 256 samples, a hop length of 64 samples, and the square root of the Hann window as the window function. A batch normalization layer is then applied, followed by a stack of bidirectional long short-term memory layers. Finally, the output of the recurrent layer is put through a fully-connected layer with a softmax activation to produce two masks which

are then applied to the complex STFT of the audio. The complex STFT is then inverted to produce individual time-domain audio sources. We use a simple 1-layer recurrent network with 50 hidden units. However, the complexity of the network can be easily scaled up to harder variants of OtoWorld.

## 4.2. Spatial Feature Extraction

The spatial features we use are the inter-aural phase and level difference between the two channels - IPD and ILD, respectively. We base our method on a simple separation method which exploits these two features to separate sources via clustering (Vincent et al., 2007; Rickard, 2007; Kim et al., 2011). The assumption of this method is that time-frequency bins that have similar spatial features are likely to come from a single direction. Sounds that come from a single direction belong to the same sources. If sources are coming from multiple directions, then you will observe two distinct clusters of spatial features.

To compute IPD, ILD the time-domain mixture audio data $x(t)$ is converted to a stereo complex spectrogram $X_{t,f}^{(c)}$ where $c$ is the channel, $t$ the time index, and $f$ the frequency index. The inter-aural phase difference and inter-aural level difference are then calculated for each time-frequency point as follows:

$$\theta_{t,f} = \angle\left(X_{t,f}^{(0)}\overline{X_{t,f}^{(1)}}\right), \tag{1}$$

$$\text{ILD}_{t,f} = \left(|X_{t,f}^{(0)}|\right)/\left(|X_{t,f}^{(1)}|\right). \tag{2}$$

While the traditional approach to exploiting IPD/ILD features is to clustering via K-Means or Gaussian Mixture Models, here our goal is to learn a monaural separation model jointly with the spatial features. The monaural separation model produces a time-frequency mask for each source $j$: $M_j(t,f)$. The mask is used to compute the mean IPD/ILD for each source:

$$\mu_j^{\text{IPD}} = \frac{1}{N}\sum_{t,f} M_j(t,f)\theta_{t,f} \tag{3}$$

$$\mu_j^{\text{ILD}} = \frac{1}{N}\sum_{t,f} M_j(t,f)\text{ILD}_{t,f} \tag{4}$$

where $N$ is the number of time-frequency points in $X(t,f)$. $\mu_j^{\text{IPD}}$ and $\mu_j^{\text{ILD}}$ are concatenated into a feature vector:

$$[\mu_0^{\text{IPD}}, \mu_0^{\text{ILD}}, \ldots, \mu_J^{\text{IPD}}, \mu_J^{\text{ILD}}] \tag{5}$$

where $J$ is the total number of sources separated by the monuaural separation network.

## 4.3. Q-Network

In our model, we use a Q-network to get the softmax probability scores over the possible action set. The spatial features

(IPD, ILD) are calculated inside of the forward pass of the Q-net and then combined with the magnitudes of STFT data along with agent information (current agent location and orientation). The combined data is passed through a fully connected layer with a PRelu activation (He et al., 2015). Finally, it is passed through a softmax layer to get the output probabilities for each possible action.

## 4.4. Loss and Optimization

We use a replay buffer and fixed-Q target networks as specified in (Mnih et al., 2015). We use the following L1 loss:

$$\mathcal{L} = |Q(s,a,w) - r + \gamma * max_{a'}Q(s',a',w_s)|_1 \tag{6}$$

Here $w_s$ are the stable weights (i.e., weights of the fixed-Q target networks). This L1 loss is back-propagated through all three model components.

We apply gradient clipping during optimization to prevent exploding gradients, and train with an Adam optimizer with an initial learning rate of 1e-2. For training, we sample randomly from the experience replay buffer with a batch size of 50. For each state that is passed into the monaural separation network, we extract 4-second excerpts.

## 5. Experiments

The purpose of this paper is to establish OtoWorld as a viable reinforcement learning environment for computer audition. We establish a baseline for a simple OtoWorld environment that can be improved upon in future works. We instantiate a very simple game in which there are two stationary sources that the agent must find. To further simplify it, the location of the sources and the agent is fixed across episodes. We use a dense reward structure, where the agent is rewarded by the lowest euclidean distance to either sources - a sort of getting hotter/colder signal. The absorption rate is set to 1.0, making the environment anechoic. With a simple, consistent environment and a dense reward structure, we find that separation models are difficult to train, suggesting that further research is needed into OtoWorld, and that OtoWorld is a challenging game due to the high complexity of representing game state as audio.

For our experiments, we used two simple audio sources: a phone ringing and a police siren. The step size was 1.0 meters and the threshold radius was 1.0 meters. If the agent did not win within 1000 steps, the episode was terminated. The first experiment ran for 50 episodes in a 6x6 meter Shoebox (rectangular) room. The second ran for 135 episodes in an 8x8 Shoebox room. Models are saved during training and could be used to test generalization in different environments.

**Baselines:** The lower baseline is a random agent that moves around the environment by sampling actions uniformly from

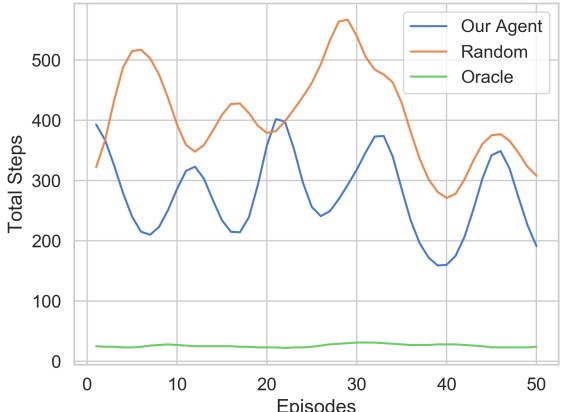

*Figure 3.* Comparing the total steps to complete an episode for our trained agent, the random agent, and the oracle agent. Our agent is significantly worse than the oracle agent but noticeably better than the random agent.

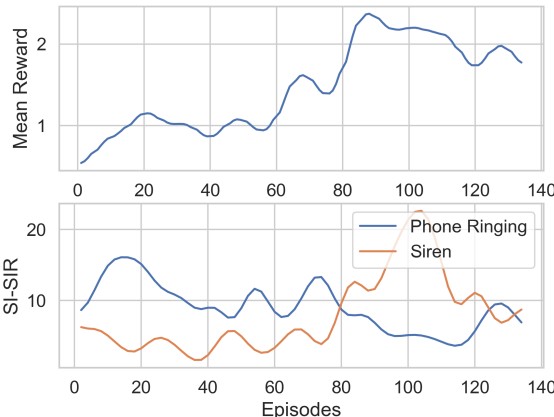

*Figure 4.* Above is the mean reward for the agent in the 8x8 Shoebox room. Below is the SI-SIR of the sources produced by the monaural separation model at each episode.

$\mathcal{A}$. This agent sometimes discovers both sources and is rewarded, depending on which actions it randomly draws. Thus, beating this agent is important to determine whether the model learns anything at all. The upper baseline is an oracle agent which knows the exact location of the sources and finishes episodes in roughly the same number of steps each time. This provides a nearly perfect benchmark for the trained agent to strive for.

**Evaluation:** We evaluate our approach in three ways: mean reward per episode, separation quality by listening, and objective separation quality using scale-invariant source-to-interference ratio (SISIR) (Le Roux et al., 2019), which measures how well an estimate matches the ground truth source. In order to obtain the ground truth, we compute the RIR at the beginning of episode with the agent and a single source in their initial locations to get the ground truth audio for that particular source.

### 5.1. Results

As shown in Figure 3, our reinforcement learning agent slightly outperforms the random agent while performing significantly worse than the oracle when it comes to how quickly the agent can complete an episode. The random agent takes on average 409 steps per episode, while the trained agent takes 271. The oracle agent far out-performs the other two, taking 26 steps on average. In Figure 4, we show the separation performance as measured by SI-SIR of the monaural separation model and its relationship with mean reward. The preliminary nature of this experiment makes it difficult to draw clear conclusions, but we observe that better separation performance can result in higher reward. However, we have noticed that separation performance in OtoWorld can be highly unstable with the mixture sometimes collapsing into one source while the other source

is empty, which occurred after episode 100 in Figure 4. Stabilizing the performance of trained models in OtoWorld will be the subject of future work.

## 6. Conclusion

OtoWorld is a challenging game for reinforcement learning agents. Indeed, our experiments are highly preliminary and separation performance and mean reward is quite low. We plan to develop OtoWorld further, as well as the agents we place into OtoWorld. In this work, we hope to have established benchmarks which researchers can strive to beat. The difficulty of OtoWorld is highly dependent on user specified settings such as the actual sources being used (whether similar or distinct), the number of sources, the size and shape of the room, and the absorption rate. As research advances, OtoWorld will be able to support more difficult and meaningful separation tasks.

OtoWorld is designed to be extended to accommodate new games and ideas. The core functionality of OtoWorld is the placement of sounding sources and agents in complex environments and an interaction paradigm where an agent can move around the environment and listen to it in different ways. While simple, to date audio-only environments for reinforcement learning are either not available, accessible, or open-source. OtoWorld seeks to fill this gap in the literature. We hope to scale up to more complex environments (e.g. speech separation), extend OtoWorld to new types of games (predator-prey games, audio-only hide-and-seek), and encourage others to use OtoWorld.

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
