# OpenReview forum: "OtoWorld: Towards Learning to Separate by Learning to Move"
_ICML.cc/2020/Workshop/SAS — SAS 2020_

### Official Review · AnonReviewer1 · 2020-06-29
**Good idea, but very confusing presentation.**

**Confidence:** 4
**Rating:** 5

**Review:**

The paper presents a framework for developing methods for acoustic based navigation. Though, the paper gives more focus on a method that is claimed to do source separation, by trying ti navigate in some conditions, in the aforementioned framework.

The idea of the method in the paper is **really** interesting, because it provides a paradigm of employing audio for navigation and in a RL paradigm. Though, the paper is not well written, there are errors in the claims and the fundamental information about the audio, and the presented results are rather not what the paper claims to be. Specifically:

1. The paper starts by trying to stress out the importance of source separation, but using wrong claims. Source separation is wrongly put as an "integral part" (as is explicitly written) of sound event detection (for example). With this wrongly established connection of computational auditory scene analysis and source separation, the paper confuses the identification and localization of a source and the actual source separation. It is really confusing, and when (around section 4) starts to be more clear, it is evident (by the type of signals) that the paper is not doing exactly what is claiming to do.

2. The paper claims that an agent can perform sound source identification and localization, because the agent can do source separation. To begin with, given the whole scientific field of audio source localization, where the people do it without using source separation, this claim is not that valid. Followingly, the paper uses a set of parameters, in a set-up similar to those used for source separation. The audio input if first processed by that set of parameters, and the output is then given to the agent in order for the agent to do localization. The paper uses 2 (only 2) sounds, and shows quite impressive values for metrics for source separation. Though, a closer look at the details can actually reveal that such a performance is not that supported from the details.

  2.1. Firstly, the two types of audio are quite different. One is a siren, the other a a phone ringing. These two sounds have considerable differences both in harmonic and temporal characteristics.

  2.2. Then, the paper does not state if the effect of the room is changed for every position of the agent. That is, the agent is moving in the room of the presented world. Does the sound stimulus change, according to the reflections and combinations of waves in the world? Are standing waves taken into account? How about the absorption coefficients of the walls? The paper says something about "energy absorption", but it does not state if this is perf frequency band.

3. It is really not clear if the paper is about the method or the world. The paper clearly states: "In this work, we present a simple game environment in which researchers can explore how to create machines that can mimic this remarkable human behavior". Then, the paper does not present any of the details of the world, for example details like the different materials that one can use, the different types of sources, the signal processing methods happening in the engine of the world, for calculating the RIR in every position, and so on.

For the above reasons, I propose the marginal rejection of the paper.

Please find below my detailed comments.

========================================================

Page 1, left column: “At the heart of computer audition is the source separation problem” that is a strong claim to make, especially for the people doing computational auditory scene analysis and not source separation. Especially with recent methods on few resources, real-life speech datasets (e.g. for infants), sound event detection, and audio captioning, this claim of the paper is not true, since source separation does not have to happen for analyzing a complex auditory scene. The paper should rephrase.

Page 1, left column: “act of isolating sound-producing sources in a mixture of sources.” at source separation, it is not the sources that are separated, but rather the audio signals that the sources are producing. Let alone, isolated, which (implies) a physical access to the source itself, which is not what usually happens, so it can be used as a definition of what is “source separation”.

Again, in the sentence “Examples include isolating a single speaker in a crowd” it is clearly implied that source separation has something to do with physical access to the source. That is not correct. What the paper (probably) wants to convey, is “speech separation”, separating an individual speech signal from a mixture of other signals.

Page 1, left column: “Source separation is an integral part of solving several computer audition tasks, such as multi-speaker speech recognition (Seki et al., 2018), music transcription and sound event detection in complex environments (Wisdom et al., 2020)” This is not true. Source separation is not an “integral part”, because there are quite many methods doing sound event detection without doing source separation. In fact, the utilization of source separation in sound event detection is quite recent, so source separation is not an “integral part”. The same goes for music transcription. The paper should find better claims for stressing out the importance of source separation.

Page 1, right column: “The current state-of-the-art for computer audition is to train deep neural networks, which learn via thousands of synthetic mixtures of isolated recordings of individual sounds.” Not true, because most current state of the art methods use real-life recorded audio and not synthetic. For example, this stands true for music source separation, acoustic scene classification, sound event detection or tagging, and audio captioning.

Page 1, right column: “As the ground truth isolated sources that go into each mixture are known, the deep network can take as input a representation of the mixture and produce estimates of the constituent sources which can be compared to the ground truth sources via a loss function”. The paper starts by referring to computer audition, but then focus solely in source separation. As shown in the previous comments, this is problematic. Also, this abstract presentation of the supervised learning for source separation, is a bit problematic because it obscures the different approaches used on source separation.

Page 1, right column: “The way that deep models learn is very different from how humans learn to separate sources. Humans rarely have sounds presented in isolation and paired with the associ- ated mixture”. This is not true.

Indeed, we learn to **identify** individual sources, but we do not do source separation. That is, as humans we learn to identify the different sources, even if these sources exist in a complex mixtures. But, when we would like to reproduce exactly what we heard, i.e. mimicking and not just saying with our own voice, then most likely we would like to hear the isolated source. Or, we hear our own voice, which is the isolated source. The paper should fix this claim.

“Instead, we learn directly from the complex mixtures that surround us every day” If the paper would like to support the claim for leaning to do source separation, the paper should present a source/citation on source separation. Now, the paper did the falsely exaggerated connection of source separation with machine listening, and now (based on this falsely exaggerated connection) the paper does wrong claims based on information that is focused on learning to detect, identify, or do anything else apart from separation.

Page 1, right column: “Part of our ability to do this stems from our ability to interact with our environment. We can move around the world, investigate sounds, and effectively change the way we hear the world by re-positioning and orienting ourselves within it.”. The paper should support these claims with proper citations.

Page 2, left column: “complex auditory environment with only sound to guide it.” the paper should clarify what is meant by “complex” here, since immediately after follows the “with only one sound”.

Page 2, left column “This game is very simple, but requires the agent to learn to separate and understand its auditory scene in order to win.” how this is verified? By “separate” is meant something specific, usually being able to provide the signal of the separated source. Here it is not mentioned anything like this. How the agent can separate sources?

Page 2, left column: “ audio source separation” it is **really** confusing how the separation is happening. In all the source separation papers, the output is the signal of the separated source. Here is never mentioned anything like this. Instead, the paper **speculates** that the joint **classification and locallization** means separation, which is wrong.

Page 2, left column: “In the former, the goal is to learn salient audio representations that are use- ful for audio tasks like event detection and source separation without access to labelled training data of any kind.” That is not true. Blind source separation is a specific sub-task of source separation and event detection is not (yet) performed in an unsupervised way (i.e. without labeled data).

Page 2, right column: “The premise is that if an agent can successfully learn to locate and turn off sources, it has implicitly learned to separate sources.” This is quite wrong and is a strong speculation, without any evidence. If an agent can “successfully learn to locate and turn off sources” then the agent surely has learnt to do source identification and localization, but it has not learned to do source separation. If the separation has to be verified, then the paper should provide typical source separation metrics, used in the source separation tasks, as mentioned in the introduction of the paper. These metrics require access to the **separated signal**.

Page 2, right column: “ energy absorption rates,” the absorption **coefficients** are per frequency bands. The paper should first use correct terminology and, then, state if such proper handling of the absorption (or reflection, i.e. 1- absorption) coefficients is used or not. The handling of the reflections in a room is a quite complex issue, with strong literature. The paper should identity its connection with the literature of room acoustics that deals with the subject of reflections in enclosures.

Page 2, right column: “Additionally, users can specify the number of sources which are then spawned at random locations in the room.”  do the locations involve height?

Page 3, left column: “Rotation is included in the action space of the agent because the orientation of the microphones can help localize sounds more accurately.” Basically, the agent utilizes only interaural level difference. The paper should state if the inter-aural time difference is also utilized, according to the set-up of the world.

Page 3, left column: At each step in an episode, the environment calculates the room impulse response (RIR) and then provides the convo- luted mixture of sources to the agent as a representation of the state of the environment.” It is not clear for which position the RIR is calculated. Does the paper and the presented world, make the modeling hypothesis that every position has the same RIR? If yes, this should be explicitly mentioned, as it is strips all the effects from corners and walls at specific distances.

Page 2, left column: “There is an option to introduce dense rewards where the agent is rewarded based on its euclidean distance to the nearest source at each step.” Is the calculation of the distance based on valid routes (i.e. not through walls) or not?

Page 4, left column: “The first experiment ran for 50 episodes in a 6x6 meter room. The second ran for 135 episodes in an 8x8 room.” Since two rooms can have same area, but very different shapes, the area alone is not enough.

---

### Official Review · AnonReviewer3 · 2020-06-29
**A reinforcement learning approach for auditory analysis**

**Rating:** 4
**Confidence:** 3

**Review:**

In this work, the authors propose a reinforcement learning-based framework for computational auditory scene analysis and source separation. Typically researchers approach the problem of source separation via supervised learning framework but this work is quite interesting due to the use of reinforcement learning. The paper is very well written and easy to follow. I have summarized my comments below which will help in improving the quality of this paper:

1. While the work uses a new framework (RL) for the auditory problem, it lacks a thorough experimental validation, and results are very mediocre. In my view, this work is very preliminary and doesn't offer much in its current shape. I would recommend the authors to extend their experiments to more challenging sounds and see if the policy is still applicable.

2.  Similar to the previous point, it would be good to see how much is the performance gap between the current state-of-the-art vs proposed approach?

3. I would like to see the generalization of the proposed approach to a variety of environments, rooms, and noise levels. Is it possible to use the same policy for rooms of different shapes and sizes? Do we need to learn the policy every single time we change the room? How does the noise level (very critical for audio applications) affect the system?

Minor comments:
Few references are missing page numbers. Please fix them.

---

### Official Review · AnonReviewer2 · 2020-06-29
**Interesting and hard RL environment for source separation**

**Rating:** 7
**Confidence:** 4

**Review:**

OtoWorld is a "navigation by listening" RL environment where in each episode the agent hears a mixture of audio from different sources in a room and it has to move to each source to turn it off. The episode is over when all sources are turned off. The hope is that the agents that excel at this task (and the more complex variants that the environment is able to construct) will excel at down-stream tasks like multi-speaker speech recognition, music transcription, and sound event detection in complex environments. This idea to my knowledge is original and few RL environments provide only-audio observations to the agent.

Pros:
* The authors provided great motivation for such an RL environment: human-like interaction and movement of the agent in the environment is an important ingredient for learning complex models of real-work audio.
* Code is publicly available.
* The paper is clearly written.
* The authors seems to have tried to train a reasonable model on a simpler case of the environment (only 2 fixed sources, etc.) and showed a lower bound (random agent) and an upper bound (oracle based on distances to each source) for possible models.

Cons:
* “The premise is that if an agent can successfully learn to locate and turn off sources, it has implicitly learned to separate sources.” --> I want to agree with this statement but the leap from "an agent that can move around and turn off sources" to a good model at down-stream tasks like multi-speaker speech recognition, music transcription, and sound event detection in complex environments is huge. This needs more explanation, or even better, comparison of models that are being trained with this environment and the models that are trained in other ways. Is there any benefit in using this environment? How do we know if there is no comparison?
* "With a simple, consistent environment and a dense reward structure, we find that separation models are difficult to train." --> I appreciate the work of the authors to train a model for this hard environment and their honesty with the less-than-ideal results. However, if it is so difficult to train an agent on the simpler form of the environment, there is a risk that research community might not want to use this environment at all. Having a reliable model that works on this configuration of the environment (or maybe even simpler) validates the  environment and provides a first step for other researchers to build upon.

---

### Decision · Program_Chairs · 2020-07-01

**Decision:**

Accept

**Comment:**

Dear author(s),

Thank you very much for your submission at the ICML2020@SaS workshop (https://icml-sas.gitlab.io/). Based on the scores assigned by the reviewers, we are happy to notify you that your paper was accepted for the workshop.

This paper has mixed reviews and we think the clarity could be improved a lot. The work is anyway relevant to the workshop and we think the idea behind this work is interesting.

Please, address the comments of the reviewers and submit the camera-ready version by July 8. We ask the authors to record a 15min video for your talk. At the workshop, we will have the pre-recorded video as well as a live QA session. It is important to keep this time limit, otherwise, your talk will be automatically cut. The deadline for uploading the video is July 8. The detailed instructions for uploading will follow.

Feel free to contact us for any questions!

Best,

The ICML20@SaS organizers:
Mirco Ravanelli
Titouan Parcollet
Dmitriy Serdyuk
Devon Hjelm
Bhuvana Ramabhadran